# Evaluating the Influenza Vaccination Knowledge Among People Living in a Rural and Medically Underserved Community of Washington State

**DOI:** 10.3390/vaccines13121233

**Published:** 2025-12-09

**Authors:** Damianne Brand, Kimberly McKeirnan, Megan Giruzzi, Juliet Dang

**Affiliations:** 1Pharmacotherapy Department, Yakima Campus, Washington State University, Yakima, WA 98901, USA; megan.giruzzi@multicare.org; 2Pharmacotherapy Department, Spokane Campus, Washington State University, Spokane, WA 99202, USA; kimberly.mckeirnan@wsu.edu; 3CSL Seqirus, North Bend, OR 97459, USA

**Keywords:** influenza vaccine, rural health, patient knowledge

## Abstract

**Background/Objectives**: Health literacy and vaccine literacy influence vaccine uptake behavior. Ensuring that people in rural communities are knowledgeable about vaccines can be an important tool in increasing influenza vaccination rates. The goal of this research was to evaluate rural community member knowledge of influenza and influenza vaccine. **Methods**: A cross-sectional survey was conducted with residents of a rural a medically underserved community in Washington State. Three thousand rural residents were contacted up to five times by a survey research center with a request to participate, with the goal of receiving 500 returned surveys based on the current population size, a z-score of 95, and an error rate of 5%. The survey evaluated rural resident knowledge and opinions about influenza and influenza vaccine. **Results**: Participants who were vaccinated against influenza in the last five years were more likely to know that influenza vaccine does not cause influenza (χ^2^ = 13.44, *p* < 0.01) and that antibiotics cannot be used to treat influenza (χ^2^ = 19.36, *p* < 0.01) than people who were not vaccinated. There was no statistical difference between people who are vaccinated and unvaccinated regarding knowing that influenza is viral rather than bacterial with the majority in both groups responding correctly (χ^2^ = 0.05, *p* < 0.82), or that people who have influenza are at higher risk for contracting pneumonia (χ^2^ = 0.78, *p* = 0.08) or COVID-19 (χ^2^ = 1.54, *p* = 0.21). Unvaccinated people were more likely to have had their opinion about vaccines changed in recent years (*p* < 0.01) and feel that COVID-19 impacted their ability to trust public health officials (*p* < 0.01). **Conclusions**: Understanding gaps that exist in rural resident knowledge about influenza could be valuable in developing future educational outreach efforts in these communities.

## 1. Introduction

Health literacy, defined as the knowledge, motivation, and capacity to find, comprehend, assess, and apply relevant health information [1], is believed to contribute to positive health outcomes by including the “informed” patient in the decision-making process [2]. Research is mixed regarding the role health literacy plays in everyday decision making and whether other factors, such as access to care, may override its importance [2,3,4,5]. Similarly, vaccine literacy and its importance on vaccine uptake behavior have become an area of interest for health care and public health providers when designing vaccine outreach and education [3]. This is especially important when addressing communities with lower overall vaccination rates, such as those in rural locations or with a high percentage of traditionally underserved populations.

Access to healthcare education or extended healthcare academic opportunities in rural areas can positively affect patient healthcare choices [6,7,8]. Many rural communities are struggling with both reduced access to healthcare and higher education, and “brain drain” as those who obtain higher levels of academic exposure tend to leave rural areas for more lucrative and expanded job opportunities offered in larger city centers and metropolitan areas [8]. This void reduces not only the spread of credible education surrounding healthcare, in general, but vaccines in particular, contributing to not only poor vaccine uptake, but the overall barrier of parsing out fact from fiction regarding current evidence-based understanding [9].

Understanding and consideration of all factors involved in patient vaccination decisions can ultimately be used to target patient educational offerings and better affect positive vaccine uptake behavior in patients. Interestingly, it has been found that health literacy is not the sole deciding factor for vaccine uptake and can be allayed by a simple recommendation from a trusted healthcare professional [5,10]. Likewise, local collective vaccination understanding, such as within a unique community, and the nature and scope of this knowledge, can help anticipate and support the patient’s understanding of influenza, the influenza vaccination, and the reduction and elimination of barriers to vaccination understanding among patients [11].

The goal of this research was to evaluate rural community member knowledge of influenza and influenza vaccine. This study was a subset of a larger survey. Further information about the study is described elsewhere [10,12,13].

## 2. Materials and Methods

### 2.1. Ethical Approval

This study was evaluated by staff from the Washington State University Institutional Review Board and found to meet the criteria for Exempt Research, meaning that full board review was not required (IRB#19877). Study procedures were conducted in accordance with university research requirements.

### 2.2. Study Design and Participants

A cross-sectional study was designed to evaluate knowledge, perspectives, attitudes, and behaviors concerning influenza vaccination among people living in a medically underserved area of Washington State. Faculty from the Washington State University Social Economic Sciences Research Center (SESRC) and the College of Pharmacy and Pharmaceutical Sciences (CPPS) collaborated to develop a survey, which was conducted in Yakima County, Washington.

Yakima County is considered a rural community by the Rural-Urban Commuting Area Codes covering over 4300 miles with approximately 60 people per square mile, compared with midsize or even denser urban areas with populations of 300 and 1000 people per square mile, respectively [14,15]. Yakima County is also designated as a medically underserved area, meaning there is a shortfall of healthcare services in the area compared with medical needs of the local population [16,17]. Opportunities for education in this community are also limited. Among residents in Yakima County, approximately 18% of the population has a bachelor’s degree compared to the state average of 40%. Yakima County is also home to one of Washington State’s largest percentages of people who identify as Hispanic or Latino/a [15]. More than half (52%) of the county’s residents are Hispanic/Latino/a, compared to an average of 14% for Washington State [15].

### 2.3. Data Collection

A 64-item survey was developed by faculty from the SESRC and CPPS to evaluate Yakima County resident opinions regarding a range of topics related to the influenza vaccine, including knowledge, perspectives, attitudes, and behaviors. Of the original 64 questions, a subset of 19 questions is included in this evaluation of rural residents’ influenza vaccination knowledge. Other survey items are reported elsewhere [10,12]. Survey questions are shown in Table 1. The survey questions were designed by the investigators with guidance from a literature search regarding current influenza vaccination knowledge and practices [18,19,20,21,22,23,24]. The SESRC also guided language clarity and question order for better survey delivery using the proprietary web survey software, DCWorks™. After development, the survey was tested by two vaccine industry experts.

The survey was conducted by the SESRC in March and April of 2023 using a five-phase administrative process [25], as shown in Table 2. The survey was distributed to 3000 random residential addresses in Yakima County with a goal of receiving a minimum of 500 participant responses, calculated based on the Cochrane formula using the current population size, a z-score of 95, and an error rate of 5%. The survey was offered as both a paper version with a stamped return envelope and a web-based survey. An introduction to the survey letter and directions on how to access the web-based survey version were mailed a week before the actual survey went out, with a $1.00 incentive to complete, followed by a postcard follow-up and a fourth survey mail-out sent to all non-respondents, and a final appeal to complete the survey provided. In each copy of the survey sent to potential participants, they were invited to consent to participate and complete the survey either by mail or online. Those who did not consent to participate could either return a response stating that they did not consent to participate or simply not respond. The survey administration period lasted approximately four weeks in total. Survey procedures for this study were designed by the SESRC based on guidance provided by the American Association of Public Opinion Research [26]. The survey was administered in English only at the recommendation of the survey research center due to previous poor response rates using surveys in other languages.

### 2.4. Data Analysis

Responses to the surveys were gathered by the SESRC and exported to CPPS researchers in a Microsoft Excel spreadsheet for analysis. Question 11 determined whether participants had been vaccinated against influenza in the previous five years. Participants were classified as “vaccinated” or “unvaccinated” based on their responses to this question. Responses to questions 12–19 were analyzed by comparing the vaccinated and unvaccinated respondents’ answers. For Questions 12–14, survey participants were asked to respond by sharing their opinion using a 4-point Likert scale, and results were analyzed using a t-test. Questions 15–19 were yes/no knowledge questions and were analyzed using a Chi-square test. Descriptive statistics (frequencies) were utilized to analyze categorical demographic variables.

## 3. Results

Out of 3000 surveys distributed, no response was received from 2318 addresses, nine were determined to be ineligible since the resident had been living in Yakima County for less than one year, and 10 responded with a refusal to participate. The survey response rate was 18.3%, as shown in Table 3. The average participant was female, married, and between the ages of 65 and 79 years old. Demographics of participants are shown in Table 4.

For Questions 12–14, survey participants were asked to respond by sharing their opinion using a 4-point Likert scale, and results between the unvaccinated and vaccinated groups were compared. Results are shown in Table 5. People who were not vaccinated against influenza were more likely to have had their opinion about vaccines changed in recent years (*p* < 0.01). People who were unvaccinated were also more likely to have felt that COVID-19 impacted their ability to trust public health officials (*p* < 0.01). There was no statistical difference between people who were vaccinated and unvaccinated regarding the impact of their political opinions on their choice about getting vaccinated. The majority of people in both groups reported their political opinions did not have an impact on their choice (*p* = 0.55).

Questions 15–19 asked participants to respond to questions showing their knowledge about influenza and the influenza vaccine as shown Table 6. People who were vaccinated were more likely to know that the flu vaccine does not cause influenza (χ^2^ = 13.44, *p* < 0.01) and that antibiotics cannot be used to treat influenza (χ^2^ = 19.36, *p* < 0.01) than people who were not vaccinated. There was no statistical difference between people who are vaccinated and unvaccinated regarding knowing that influenza is viral rather than bacterial, with the majority in both groups responding correctly (χ^2^ = 0.05, *p* < 0.82). Participants in both groups were less likely to know that people who have influenza are more likely to get pneumonia (χ^2^ = 0.78, *p* = 0.08) or COVID-19 (χ^2^ = 1.54, *p* = 0.21). There was no statistically significant difference between groups for either question.

## 4. Discussion

This research sought to evaluate rural community member knowledge and opinions about influenza and influenza vaccines. Understanding whether gaps exist in rural resident knowledge about influenza could be valuable in developing future educational outreach efforts in these communities. Rural residents experience limited access to healthcare services and can have vaccination rates 40% lower than urban areas [27,28]. Unvaccinated survey participants were more likely to have experienced changes in their opinions about vaccines in recent years and felt that the pandemic impacted their ability to trust public health officials. These results were similar to previous work, identifying that the influence of politics and a lack of trust in the government were barriers to vaccination uptake during the COVID-19 pandemic [29] were substantially factors in rural communities. Rebuilding trust is critical because one of the most effective strategies for encouraging vaccination is a strong recommendation from a trusted healthcare provider [30,31,32].

As demonstrated by this survey, many misconceptions are circulating in the community about influenza and the influenza vaccines. The majority of participants in both the vaccinated and unvaccinated groups knew that influenza is caused by a virus. More people in the vaccinated group knew that antibiotics do not treat influenza. This may be a confusing topic for laypersons because direct-to-patient marketing for antiviral medications may lead to mistaking these influenza treatments with antibiotics. Many individuals also did not understand the efficacy of the flu vaccine or the risk of concomitant pneumonia. This could be due to many factors, including gaps in knowledge, circulation of misinformation, and limited health literacy.

Additional misconceptions among survey participants were identified. More than 40% of individuals who answered this survey believed that you could contract the flu from the flu shot or were unsure. This may be a confusing topic for patients, in part because the live attenuated influenza vaccine (LAIV) can lead to a positive influenza test even though the patient cannot contract influenza from it. We believe this is less possible as the LAIV was not likely used in our population due to the age of our respondents. A Cochrane review concluded that the inactivated influenza vaccine reduced the incidence of positive influenza tests, without causing the disease itself [33]. Educating patients on how the influenza vaccine works could help correct this myth. Education could be as simple as the following: The influenza vaccine is made from an inactivated virus and therefore cannot transmit an active infection. The vaccine works by telling the body to produce antibodies to the inactivated virus to build up a defense system. It can take two weeks for the body to create sufficient antibodies. Therefore, patients who contract the flu or get sick around the same time as the influenza vaccine were already going to get sick.

Preventing influenza and other vaccine-preventable diseases is of great importance for health providers serving rural communities. Concerted outreach efforts to promote vaccine accessibility and education have already helped increase vaccination rates [34,35]. However, more can be done to affect vaccination uptake among rural patient populations. To improve this outreach, providers need to continue to better understand the patient’s knowledge of influenza as a preventable disease, how it can contribute to other potential infectious diseases, and the role vaccines play in their ultimate care [36]. There continues to be a misconception among certain patient populations of the real dangers of influenza infections, not only to the medically vulnerable, but to pregnant people and, in most years, the elderly and prepubescent [37]. Some of this confusion may be perpetuated by variances in vaccine effectiveness season-to-season to completely stop influenza, and also the lack of understanding that the influenza vaccine can not only be used to reduce the severity of illness but also protect against susceptibility to other infectious diseases [38]. This can inform providers to better meet patients where they are and further mitigate these barriers or promote continued access strategies with proven success [39].

There are limitations to this cross-sectional study, which was conducted in one rural area of one state. Results may not be generalizable to rural residents in other states or countries. Although the study survey was developed in collaboration with a survey research center and piloted by two vaccine experts, this survey was not validated. Selection bias is also likely since survey participants chose to complete the survey. People who are uninterested in vaccination or have negative viewpoints about vaccination may have been less likely to respond, so results may underrepresent the opinions of people who are unvaccinated. The survey was only administered in English and may not have captured the perspectives of rural community residents who did not speak English. This language limitation may have led to a selection bias, as 52% of the population of Yakima County identifies as Hispanic/Latino/a, and their comfort level with the English language was not assessed. Additionally, this research asked participants about political opinions but did not ask them to identify their political party. Additional research evaluating differences between political parties may be valuable.

Further outreach designed to increase rural patient knowledge about influenza and influenza vaccine could increase patient health literacy. Rural residents may also be more invested in their health if they are more aware of the risks influenza poses to them and those they care about [40]. Better awareness and development of vaccine literacy can help to recognize it as a contributor to their vaccination hesitancy.

## 5. Conclusions

Understanding gaps that exist in rural resident knowledge about influenza could be valuable in developing future educational outreach efforts in these communities. People who were not vaccinated against influenza were more likely to have had their opinion about vaccines changed in recent years, to have felt that the pandemic impacted their ability to trust public health officials, and were less likely to be knowledgeable about influenza vaccine than those who were vaccinated. Providing education so that people in rural communities are knowledgeable about influenza and the influenza vaccine could be an important tool in increasing influenza vaccination rates.

## Figures and Tables

**Table 1 vaccines-13-01233-t001:** Study survey questions.

Question Number	Question	Topic Domain
1	Have you lived in Yakima County for the past 12 months or longer?	Inclusion criteria
2	Are you age 18 or older?	Inclusion criteria
3	Which of the following is your age group?	Demographics
4	Please indicate your race: white or Caucasian; Black or African American; Asian; Native Hawaiian or Other Pacific Islander; American Indian/Alaska Native/Tribal; Other, please specify.	Demographics
5	Please indicate your ethnicity: Latino/Latina//Hispanic; Non-Hispanic.	Demographics
6	Which gender do you identify as?	Demographics
7	What is the highest level of education you have completed: less than high school; high school or GED; some college/AA degree/technical certificate; bachelors degree; advanced degree (PhD, Masters, MD, etc.); other, please specify.	Demographics
8	Which one of the following best describes your current marital status?	Demographics
9	Which of the following categories best describes your current employment status?	Demographics
10	Have you ever served in any branch of the United States military?	Demographics
11	How often in the past five years (2018–2022) have you been vaccinated for the flu?	Influenza vaccination status
12	Has your view/opinion about vaccines changed in recent years?	Opinion about influenza vaccination
13	Has COVID-19 impacted your ability to trust public health officials?	
14	Do your political opinions impact your choice about getting vaccinated?	
15	Can you get the flu from the flu shot?	Knowledge of influenza vaccine
16	Is the flu viral or bacterial?	
17	Can antibiotics be used to treat the flu?	
18	Are people who get the flu at higher risk for getting pneumonia?	
19	Are people who get the flu at higher risk for getting COVID-19?	

Abbreviations used: GED: General Educational Development (U.S. high school equivalency degree); AA: Associate of Arts degree; PhD: Doctor of Philosophy degree; MD: Medical Doctor degree.

**Table 2 vaccines-13-01233-t002:** Data collection timeline.

Scheme 3	Date	Number Sent
Pre-notification Letter	27 March 2023	3000
Letter with questionnaire	3 April 2023	3000
Reminder postcard	11 April 2023	2774
Replacement questionnaire	19 April 2023	2732
Final reminder letter	28 April 2023	2491

**Table 3 vaccines-13-01233-t003:** Survey sample disposition.

Survey Response	Number of Surveys
Survey completed online	221
Survey completed by mail	279
Survey partially completed online	20
Refusal to participate	10
No response	2318
Survey undeliverable using address provided	143
Responded ineligible (living in Yakima County for less than 12 months)	9
Total	3000

**Table 4 vaccines-13-01233-t004:** Survey respondent demographics.

Demographic	Number	Percentage
Race (*n* = 485) *		
American Indian/Alaska Native/Tribal	19	4%
Asian	10	2%
Black or African American	8	2%
Native Hawaiian or Other Pacific Islander	3	1%
Other	26	5%
White or Caucasian	419	86%
Ethnicity (*n* = 461) *		
Latino/Latina/Hispanic	78	17%
Non-Hispanic	383	83%
Highest level of education (*n* = 484) *		
Less than high school	17	4%
High school or GED	104	21%
Some college/AA degree/technical certificate	161	33%
Bachelor’s degree	112	23%
Advanced degree (Masters, PhD, MD, etc.)	87	18%
Other	3	1%
Employment status (*n* = 487) *		
Employed full-time	191	39%
Employed part-time	25	5%
Not employed, looking for employment	13	3%
Not employed, not looking for employment	25	5%
Retired	233	48%
Ever served in the United States military (*n* = 493) *		
Yes, now on active duty	1	<1%
Yes, but not on active duty in last 12 months	50	10%
No, but trained for Reserves or National Guard	4	1%
No, never served in the military	438	88%

* Individual survey questions have varying response rates because participants could choose to skip questions. Abbreviations used: GED: General Educational Development (U.S. high school equivalency degree); AA: Associate of Arts degree; PhD: Doctor of Philosophy degree; MD: Medical Doctor degree.

**Table 5 vaccines-13-01233-t005:** Survey participant responses to opinion questions.

Question	Population *	Strongly (4)	Mostly (3)	Only Slightly (2)	Not at all (1)	*p*-Value
12. Has your view/opinion about vaccines changed in recent years?	Unvaccinated (*n* = 130)	26 (20%)	18 (14%)	30 (23%)	56 (43%)	<0.01
	Vaccinated (*n* = 352)	35 (10%)	38 (11%)	106 (30%)	173 (49%)	
13. Has COVID-19 impacted your ability to trust public health officials?	Unvaccinated (*n* = 133)	62 (46%)	18 (14%)	18 (14%)	35 (26%)	<0.01
	Vaccinated (*n* = 354)	87 (25%)	51 (14%)	92 (26%)	124 (35%)	
14. Do your political opinions impact your choice about getting vaccinated?	Unvaccinated (*n* = 131)	5 (4%)	10 (8%)	29 (22%)	87 (66%)	0.55
	Vaccinated (*n* = 353)	23 (7%)	18 (5%)	49 (14%)	263 (74%)	

* Survey participants classified as “unvaccinated” or “vaccinated” against influenza based on response to Question 11.

**Table 6 vaccines-13-01233-t006:** Survey participant responses to knowledge questions.

Question	Population	Answered Correctly *n* (%)	Answered Incorrectly or Not Sure *n* (%)	Chi-Squared Likelihood Ratio	df	*p*-Value
15. Can you get the flu from the flu shot?	Unvaccinated (*n* = 126)	64 (51%)	62 (49%)	13.44	1	<0.01
	Vaccinated (*n* = 355)	245 (69%)	110 (31%)			
16. Is the flu viral or bacterial?	Unvaccinated (*n* = 125)	100 (80%)	25 (20%)	0.05	1	0.82
	Vaccinated (*n* = 353)	279 (79%)	74 (21%)			
17. Can antibiotics be used to treat the flu?	Unvaccinated (*n* = 126)	93 (76%)	30 (24%)	19.36	1	<0.01
	Vaccinated (*n* = 357)	325 (91%)	32 (9%)			
18. Are people who get the flu at higher risk for getting pneumonia?	Unvaccinated (*n* = 127)	67 (53%)	60 (47%)	0.08	1	0.78
	Vaccinated (*n* = 358)	194 (54%)	164 (46%)			
19. Are people who get the flu at higher risk for getting COVID-19?	Unvaccinated (*n* = 126)	38 (30%)	88 (70%)	1.54	1	0.21
	Vaccinated (*n* = 355)	87 (25%)	268 (75%)			

Abbreviations used: df: degrees of freedom.

## Data Availability

The datasets presented in this article are not readily available because the data are part of an ongoing study. Requests to access the datasets should be directed to the corresponding author.

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
