# Peer review of "Evaluating the Influenza Vaccination Knowledge Among People Living in a Rural and Medically Underserved Community of Washington State"

_vaccines, 2025, doi:10.3390/vaccines13121233_

Round 1

Reviewer 1 Report

Comments and Suggestions for Authors

I read with interest your manuscript attempting to identify factors/knowledge associated with influenza vaccination in a rural county in Washington state.

I have a few comments and questions that I would like to have answered.

1. You developed a 64-item survey yet only report 19 questions. Were all of the questions administered and you are only reporting these 19 or was this the entire survey? Can you add to the description of the study to indicate what the time frame of the completed survey was? I think this is important for readers to understand the context related to the COVID-19 pandemic or political environment of country.

2. You reported that 50% of the county identifies as Hispanic/Latino. Was the survey administered in both English and Spanish? Were any other languages offered? I think this is extremely important as your survey results note that only 17% of your respondents identified in this way. I believe that you must address this shortfall in your discussion and provide possible explanations for this major deviation of the demographics. You suggest that this may not be generalizable to other rural populations but may not even represent your population.

3. You asked whether your political opinions impact your decision making but did you include a question about political identification in your demographics question that was not included in the study. I would be interested by this fact as it might show hidden biases that were not expressed overtly.

4. A few questions regarding your survey questions asked - Can you comment on these?

A. Can antibiotics be used to treat the flu? This could be misleading. I am not sure that antibiotics for bacteria and antivirals for influenza are easily discriminated by lay persons. Direct to patient marketing of oseltamivir and other newer antivirals may be used to treat an infection.

B. Your description of the education plan on line 189 is a bit faulty. Flumist is one version of the influenza vaccine for patients 5-49 years of age. It is a live attenuated vaccine and not an inactivated virus as stated in line 190 and can in fact lead to a positive PCR or antigen test.

5. Was willingness to be immunized against other pathogens assessed? Did you ask if patients received other vaccines or are patients not receiving the the flu vaccine simple not receiving any vaccines? I would think that these survey respondents may want to be excluded since their refusal is not related to influenza but all vaccines.

Reviewer 2 Report

Comments and Suggestions for Authors

Overall the paper was well-written and interesting. There are a couple of issues that need to be addressed/explained.  

Section 2: Materials and Methods: You write that the study met the criteria for Exempt Research. Does this mean that it was exempt from full board review or exempt from having to secure Informed Consent (IC) from the participants? I cannot find where you received IC. Please explain. 

Has your survey been assessed for validity? Please report on this one way or the other. 

Comments on the Quality of English Language

The quality of the English is fine.  

Reviewer 3 Report

Comments and Suggestions for Authors

This is an investigative study on the attitudes towards influenza vaccines in a single  community. It has certain promotional and advisory effects for local strategies to improve the vaccination rate of influenza vaccines, especially during the period when vaccine hesitancy become a significant challenge in vaccine administration. Although this study is a single- community research, it also has reference value for promoting the vaccination rate of influenza vaccines in other regions.

In my opinion, several aspects of the manuscript require revision prior to official publication. Firstly, in the results section, including Table 3, states that a total of 3,000 cases were included as study participants; however, the reported number of 2,848 cases and the associated figure of 18.3% are inconsistent or unclear, which may hinder reader comprehension. A more precise and transparent explanation is required to ensure clarity and data coherence.

In Table 5, clarification is required regarding which two datasets were compared to derive the P value.

And in the discussion section, certain arguments appear inconsistent with the conclusions supported by the study's results. Specifically, the increase of trained vaccinators and concerted outreach efforts to promote vaccine accessibility and education,have served to increase vaccination rates in a profound way, however, this finding is not fully reflected or adequately discussed in this study.

In the conclusion, the results of the research were not presented.

Round 2

Reviewer 1 Report

Comments and Suggestions for Authors

Thank you for your consideration of the comments made in the initial review. I have a few additional adjustments that I would recommend for clarity and transparency.

Line 200:

Due to the fact that your average respondent was between 65-75 years of age, the impact of the LAIV (Flumist) is likely to be less. Therefore, I think you could add a line after the sentence "This may be a confusing topic for patients, in part be-199 cause live attenuated influenza vaccine can lead to a positive influenza test even though 200 the patient cannot contract influenza from it." We believe this is less likely as the LAIV was not likely used in our population due to the age of our respondents. 

Line 233 - After you note that the limitation related to the survey being English language only, you should consider adding "This language limitation may have led to selection bias as 52% of the population of Yakima County identifies as Hispanic/Latino/a and their comfortability with the English language was not assessed."

Otherwise, thank you for addressing my concerns to the best of your ability. 

Author Response

Revisions added in Green highlight on revised version of paper on second review:

Comment 1: Due to the fact that your average respondent was between 65-75 years of age, the impact of the LAIV (Flumist) is likely to be less. Therefore, I think you could add a line after the sentence "This may be a confusing topic for patients, in part be-199 cause live attenuated influenza vaccine can lead to a positive influenza test even though 200 the patient cannot contract influenza from it." We believe this is less likely as the LAIV was not likely used in our population due to the age of our respondents.  YES. Accept revision suggestion.  We agree it was important to clarify the impact of this response with age related exposure to the LAIV and subsequent positive test results.

Comment 2: Line 233 - After you note that the limitation related to the survey being English language only, you should consider adding "This language limitation may have led to selection bias as 52% of the population of Yakima County identifies as Hispanic/Latino/a and their comfortability with the English language was not assessed." YES Accept revision suggestion.  The potential bias associated with language barriers was considered but forgone due to suggestions through the survey group as they have shown historically minimal variance in offering a Spanish version in these populations with a reliable high return rate for older white women regardless.  The additional statement can add clarity on which population was measured to help understand possible impact of results.